# Clustered functional domains for curves and corners in cortical area V4

**Rundong Jiang[1,2,3,4], Ian Max Andolina[5], Ming Li[6], Shiming Tang[1,2,3,4]\***

[1]Peking University School of Life Sciences, Beijing, China; [2]Peking-Tsinghua Center for Life Sciences, Beijing, China; [3]IDG/McGovern Institute for Brain Research at Peking University, Beijing, China; [4]Key Laboratory of Machine Perception (Ministry of Education), Peking University, Beijing, China; [5]The Center for Excellence in Brain Science and Intelligence Technology, State Key Laboratory of Neuroscience, Key Laboratory of Primate Neurobiology, Institute of Neuroscience, Chinese Academy of Sciences, Shanghai, China; [6]Beijing Normal University Faculty of Psychology, Beijing, China

**Abstract** The ventral visual pathway is crucially involved in integrating low-level visual features into complex representations for objects and scenes. At an intermediate stage of the ventral visual pathway, V4 plays a crucial role in supporting this transformation. Many V4 neurons are selective for shape segments like curves and corners; however, it remains unclear whether these neurons are organized into clustered functional domains, a structural motif common across other visual cortices. Using two-photon calcium imaging in awake macaques, we confirmed and localized cortical domains selective for curves or corners in V4. Single-cell resolution imaging confirmed that curve- or corner-selective neurons were spatially clustered into such domains. When tested with hexagonal-segment stimuli, we find that stimulus smoothness is the cardinal difference between curve and corner selectivity in V4. Combining cortical population responses with single-neuron analysis, our results reveal that curves and corners are encoded by neurons clustered into functional domains in V4. This functionally specific population architecture bridges the gap between the early and late cortices of the ventral pathway and may serve to facilitate complex object recognition.

**\*For correspondence:** tangshm@pku.edu.cn

**Competing interests:** The authors declare that no competing interests exist.

## Introduction

The visual system faces the daunting task of combining highly ambiguous local patterns of contrast into robust, coherent, and spatially extensive complex object representations (*Connor et al., 2007*; *Haxby et al., 1991*; *Mishkin et al., 1983*). Such information is predominantly processed along the ventral visual pathway (areas V1, V2, V4, and inferotemporal cortex [IT]). At early stages of this cortical pathway, neurons are tuned to a local single orientation (*Hubel and Livingstone, 1987*; *Hubel and Wiesel, 1968*) or a combination of orientations (*Anzai et al., 2007*; *Ito and Komatsu, 2004*). Orientation responses are functionally organized into iso-orientation domains that form pinwheel structures in V1 (*Ts'o et al., 1990*). At later stages like IT, neurons are selective for complex objects, predominantly organized categorically (*Desimone et al., 1984*; *Freiwald and Tsao, 2010*; *Fujita et al., 1992*; *Kobatake and Tanaka, 1994*; *Tsao et al., 2003*; *Tsao et al., 2006*). Such complex object organization is embodied using combinations of structurally separated feature columns (*Fujita et al., 1992*; *Rajalingham and DiCarlo, 2019*; *Tanaka, 2003*; *Tsunoda et al., 2001*; *Wang et al., 1996*). Positioned in-between the local orientation architecture of V1 and the global object architecture of IT lies cortical area V4, exhibiting visual selectivity that demonstrates integration of simple-towards-complex information (*Pasupathy et al., 2019*; *Roe et al., 2012*; *Yue et al.,*

2014), and extensive anatomical connectivity across the visual hierarchy (*Gattass et al., 1990*; *Ungerleider et al., 2008*).

Functional organization within V4 has previously been visualized by intrinsic signal optical imaging (ISOI), and cortical representations of low-level features for orientation, color, and spatial frequency have been systematically demonstrated (*Conway et al., 2007*; *Li et al., 2014*; *Li et al., 2013*; *Lu et al., 2018*; *Tanigawa et al., 2010*). Such functional clustering suggests that the intracortical organizational motifs in V4 bear some similarity to V1. It remains unknown how more complex feature-selective neurons in V4 are spatially organized, and whether feature-like columns found in IT also exist in V4. Because intrinsic imaging is both spatially and temporally limited, it is unable to measure selective responses of single neurons. Using electrophysiology, early studies in V4 using bar and grating stimuli found that V4 neurons are tuned for orientation, size, and spatial frequency (*Desimone and Schein, 1987*). Subsequent studies revealed V4 selectivity for complex gratings and shapes in natural scenes (*David et al., 2006*; *Gallant et al., 1993*; *Kobatake and Tanaka, 1994*). In particular, Gallant and colleagues discovered V4 neurons with significant preferences for concentric, radial, and hyperbolic gratings (*Gallant et al., 1993*; *Gallant et al., 1996*). Neurons with similar preferences were spatially clustered when reconstructing the electrophysiological electrode penetrations (*Gallant et al., 1996*). These results were extended by later studies confirming the systematic tuning of V4 neurons for shape segments such as curves and corners as well as combination of these segments using parametric stimulus sets consisting of complex shape features (*Cadieu et al., 2007*; *Carlson et al., 2011*; *Oleskiw et al., 2014*; *Pasupathy and Connor, 1999*; *Pasupathy and Connor, 2001*; *Pasupathy and Connor, 2002*). Temporally varying heterogeneous fine-scale tuning within the spatial-temporal receptive field has also been observed (*Nandy et al., 2016*; *Nandy et al., 2013*; *Yau et al., 2013*). More recently, artificial neural networks were used to generate complex stimuli that characterize the selectivity of V4 neurons (*Bashivan et al., 2019*). However, whether such complex feature-selective neurons are spatially organized in V4 remains poorly understood.

In this study, we aimed to confirm the presence of functional domains in V4 encoding complex features such as curves and corners. We utilized two-photon (2P) calcium imaging in awake macaque V4, which provides visualization of the spatial distribution and clustering within the cortical population alongside substantially enhanced spatial resolution for functional characterization at the single-cell level (*Garg et al., 2019*; *Li et al., 2017*; *Nauhaus et al., 2012*; *Ohki et al., 2005*; *Seidemann et al., 2016*; *Tang et al., 2018*). We scanned a large cortical area in dorsal V4 using a low-power objective lens to search for patches selectively activated by curves or corners. We subsequently imaged these patches using a high-power objective lens to record single neurons' responses in order to examine whether spatially clustered curve or corner-selective neurons could be found. If such neural clusters were found, we further aimed to understand how different curves and corners are encoded and differentiated in greater detail.

## Results

We injected AAV1-hSyn-GCaMP into dorsal V4 (V4d) of two rhesus macaques — GCaMP6f for monkey A and GCaMP5G for monkey B. An imaging window and head posts were implanted 1–2 months after viral injection (see Materials and methods). Subjects were trained to initiate and maintain fixation within a 1° circular window for 2 s: the first second contained the fixation spot alone, and then stimuli appeared for 1 s on a LCD monitor positioned 45 cm away (17 inch, 1280 × 960 pixel, 30 pixel/°). Neuronal responses were recorded using 2P calcium imaging, with differential images generated using $\Delta F = F - F0$, where F0 is the average fluorescence 0.5–0 s before stimulus onset, and F is the average response 0.5–1.25 s after stimulus onset.

### Cortical mapping of curve-biased and corner-biased patches in V4

We first identified the retinal eccentricity using drifting gratings for our sites and found they were positioned with an eccentricity of ~0.7° from the fovea in monkey A and ~0.9° in monkey B. We next used a low-power (4×) objective lens to identify and localize any cortical subregions selectively activated by curves or corners. Using a large range of contour feature stimuli including bars, curves, and corners (*Figure 1A*), we scanned a large area (3.4 × 3.4 mm) in V4d (*Figure 1B, C*, *Figure 1—figure supplement 1*) between the lunate sulcus (LS) and the terminal portion of the inferior occipital sulcus (IOS). We obtained global activation maps by Gaussian smoothing (standard deviation σ = 10 pixels,

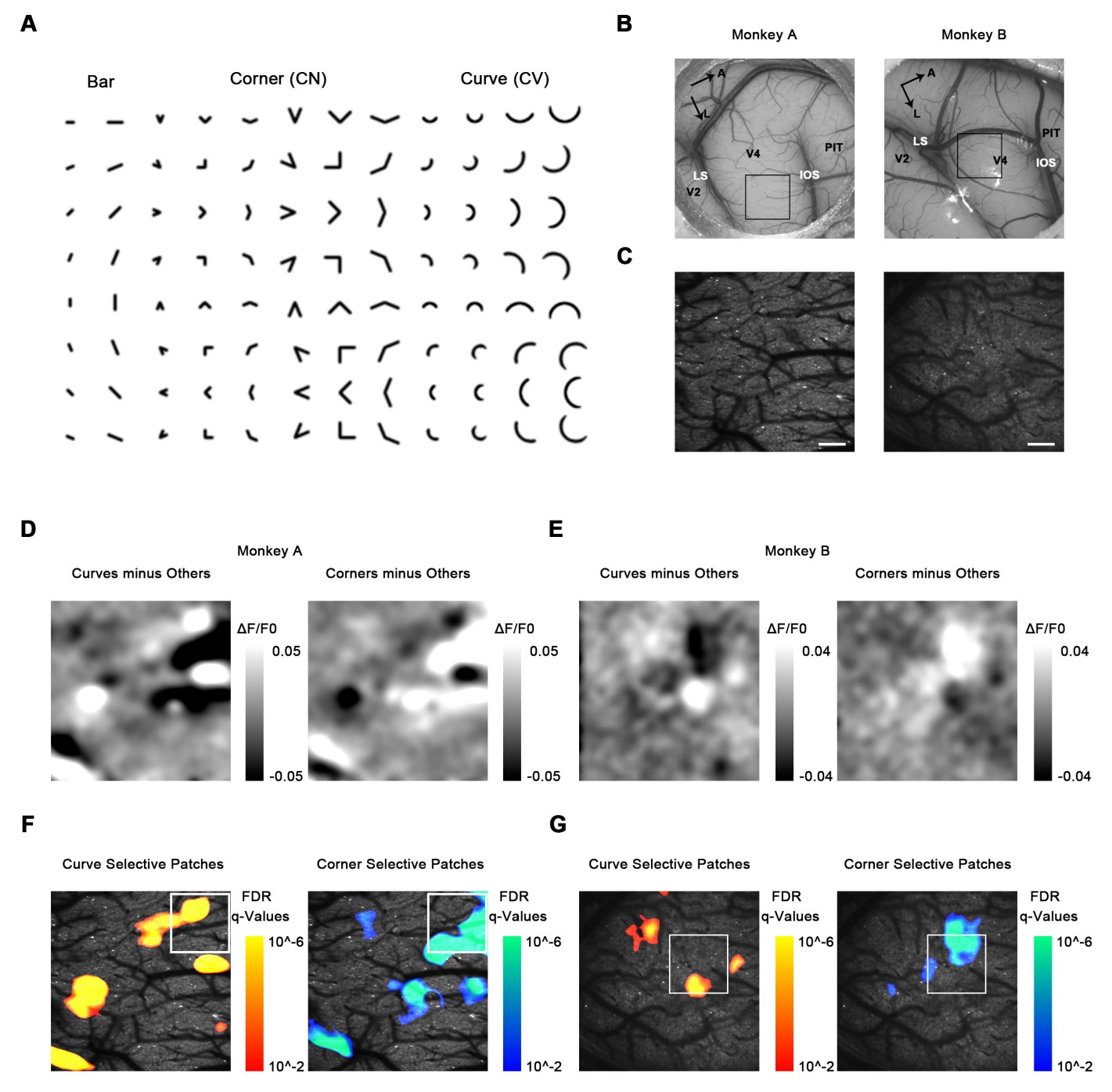

**Figure 1.** Cortical mapping of curve-biased and corner-biased patches in V4 using a 4× objective lens. (**A**) The stimulus set used for initial cortical mapping consisting of bars, corners, and smooth curves. (**B**) Vascular map. LS: lunate sulcus; IOS: inferior occipital sulcus. The black box indicates the imaging site in each subject. (**C**) Two-photon fluorescence images of the two monkeys. Scale bar = 400 μm. (**D**) Left: subtraction map showing curve-selective activation in monkey A, derived by the average response (ΔF/F0) to all curves minus the average response to all other stimuli (corners and bars). Right: subtraction map showing corner-selective activation in monkey A. (**E**) The equivalent of (**D**) for monkey B. (**F**) Left: significant curve patches in monkey A. For each pixel, independent t-tests were performed to compare the responses to all curves against all corners and against all bars. Benjamini-Hochberg procedure was used to compute the pixel FDR (false discovery rate, see Materials and methods). Threshold q = 0.01. The white box indicates the imaging site selected for 16× objective single-cell mapping. Right: significant corner patches in monkey A. (**G**) The equivalent of (**F**) for monkey B.

The online version of this article includes the following figure supplement(s) for figure 1:

*Figure 1 continued on next page*

*Figure 1 continued*

**Figure supplement 1.** Two-photon fluorescence images.

**Figure supplement 2.** Pseudo-color orientation map obtained by 4× imaging.

**Figure supplement 3.** Maps of uncorrected p-values (p<0.01), FDR q-values (q < 0.01), and Bonferroni-corrected p-values (p<0.01), before cluster permutation tests.

67 µm) the ΔF/F0 maps. We observed that orientation is organized in linear iso-orientation domains or pinwheel-like patterns, as previously reported (*Roe et al., 2012*), using ISOI in V4 (*Figure 1—figure supplement 2*).

We then examined the response to curve and corner stimuli. Using map subtraction, we computed the curve-selective activation as the average response (ΔF/F0) to all curves minus all corners and bars, and corner-selective activation as the average response to all corners minus all curves and bars (*Figure 1D, E*). The subtraction maps we obtained clearly revealed several possible candidates for curve- or corner-selective patches. To statistically detect and locate the curve and corner patches, we performed pixel-level FDR tests to examine the curve or corner preference. For each pixel, we performed independent t-tests to compare the responses to all curves, all corners, and all bars, obtaining the p-value maps for curve and corner selectivity (see Materials and methods and *Figure 1—figure supplement 3*). We then computed the FDR (false dicovery rate) using Benjamini-Hochberg procedure (*Benjamini and Hochberg, 1995*), with the threshold level q = 0.01 to locate the significant patches. Cluster permutation tests were also performed to exclude patches with not enough significant pixels (*Nichols and Holmes, 2002*). We found several patches significantly selective to curves or corners in dorsal V4 (*Figure 1F, G*). These curve- or corner-selective patches were considered candidates for functional domains encoding shape segments in V4.

## Single-cell mapping of curve- and corner-selective neurons reveals they are spatially clustered

To confirm that neurons within these patches were indeed curve or corner selective, we next performed single-cell resolution imaging with a high-power objective lens (16×) to record neuronal responses (ΔF/F0) as well as their spatial organization (*Figure 2—figure supplement 1*). The imaging sites (850 × 850 µm) in both subjects were chosen to include both curve- and corner-selective domains found by our 4× imaging (*Figure 1F, G*). 535 visually responsive neurons (292 from monkey A and 243 from monkey B) were recorded in total. Each stimulus was repeated 10 times and averaged to derive neuronal responses (*Figure 2—figure supplement 2*). To characterize neurons' curve and corner selectivity, we calculated a curve selectivity index (CVSI) and corner selectivity index (CNSI). A positive CVSI value indicates a neuron's maximal response to curves is stronger than its maximal response to other stimuli: a CVSI = 0.33 signifies a response twice as strong, and a CVSI = 0.2 is 1.5 times as strong. The same definition applies to CNSI. 70.5% (74 out of 105) neurons with CVSI > 0.2 significantly (one-way ANOVA, p<0.05) preferred curves over corners and bars, and 76.9% (120 out of 156) for CNSI (*Figure 2—figure supplement 3A, B*). We found neurons with high CVSI or CNSI were spatially clustered (*Figure 2A–D*), and these neurons were also selective to the orientation of the integral curves or corners (*Figure 2E–H*; 91.6% of the neurons are significantly tuned to the orientation of curves or corners; one-way ANOVA, p<0.05). Their overall spatial distribution was consistent with the spatial distribution of curve and corner domains revealed by 4× imaging (*Figure 2A–D* vs. *Figure 1E, F*), especially considering the possible loss of detailed spatial information during Gaussian smoothing of 4× images. This parsimoniously suggests that the observed cortical activation was evoked by responsive neuronal clusters.

We next assessed this clustering quantitatively by examining how neuronal responses correlate with spatial distance. For each neuronal pair recorded from the same subject, we computed the pairwise tuning correlation and absolute value differences for CVSI and CNSI plotted against the neuronal pairwise distances. We found that neurons close to each other (<300 µm approximately) often had more correlated tuning (*Figure 2I*) and generally exhibited more similar CVSI and CNSI values (*Figure 2J, K*). These results indicate curve-selective and corner-selective neurons are spatially clustered, which could potentially form curve domains and corner domains in V4, which could therefore be detected when imaged at a larger scale.

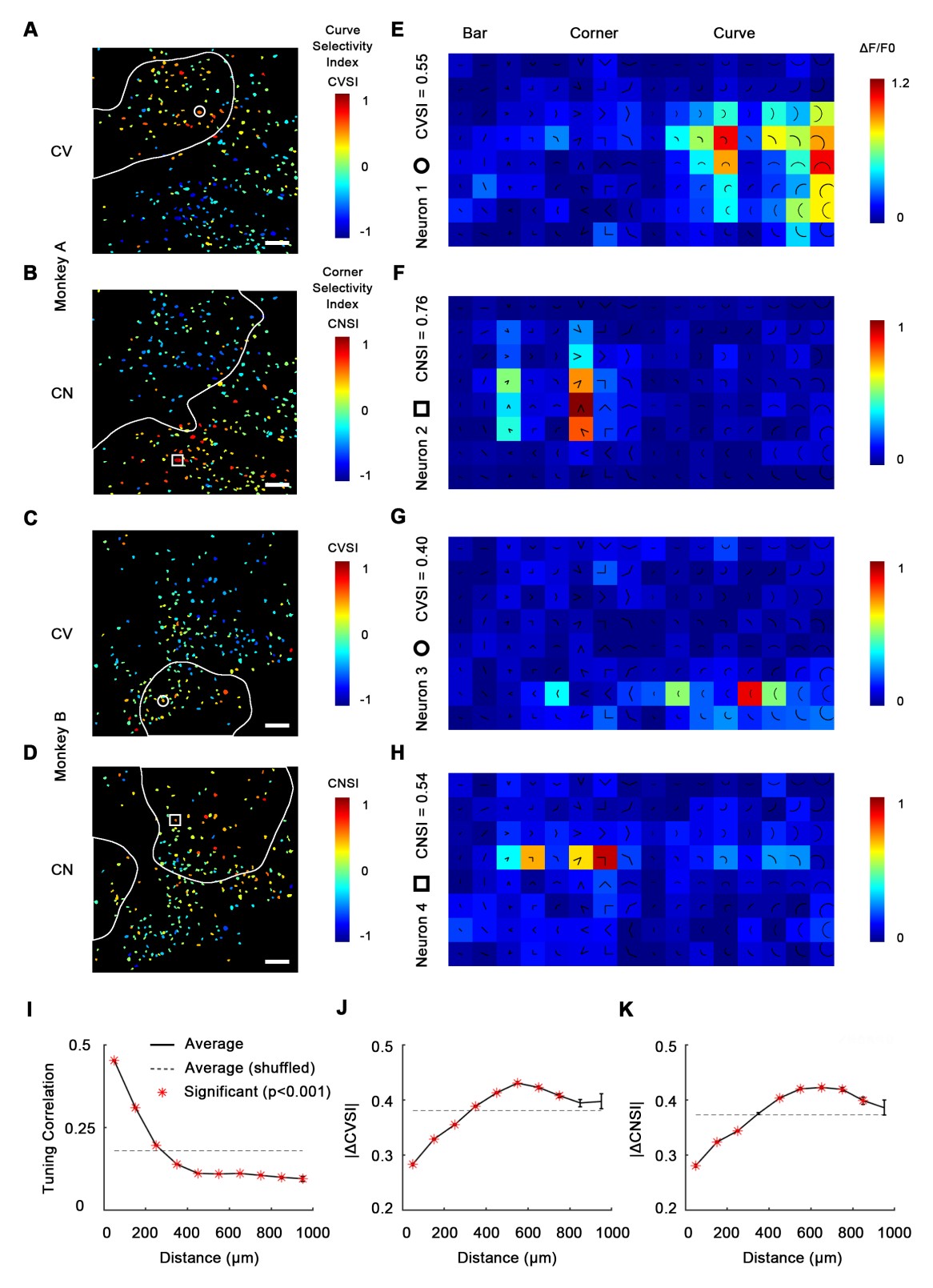

**Figure 2.** Single-cell mapping of curve- and corner-selective neurons using a 16× objective lens. (A) Cell map of curve selectivity index (CVSI). Responsive neurons are labeled at their spatial location and colored according to their CVSI. Neurons with high positive CVSI (high curve preference) were clustered in the upper part of the imaging area. The white line indicates the curve-biased patches derived by 4× imaging (*Figure 1E*). Scale bar = 100 µm. (B) Cell map of corner selectivity index (CNSI). Neurons with high positive CNSI (high corner preference) were clustered in the lower part

*Figure 2 continued on next page*

*Figure 2 continued*

of the imaging area. (**C, D**) Equivalent maps for monkey B. (**E–H**) Responses of four example neurons preferring curves or corners, their locations labeled in (**A–D**), respectively. (**I**) Neuronal pairwise tuning correlation (mean ± SE, averaging all neurons every 100 µm) plotted against spatial distances. The average correlation between different repeats of same neuron is 0.71 (*Figure 2—figure supplement 2*). The dash curve indicates the average of neurons when shuffled. Significance levels were determined by permutation test. (**J**) Absolute CVSI value differences (mean ± SE) plotted against distances. (**K**) Absolute CNSI value differences (mean ± SE) plotted against distances.

The online version of this article includes the following figure supplement(s) for figure 2:

**Figure supplement 1.** Single-cell resolution fluorecence imge.

**Figure supplement 2.** Single-neuron responses.

**Figure supplement 3.** Curve selectivity index (CVSI) and corner selectivity index (CNSI).

Out of all 535 neurons recorded from two animals, the majority (346 neurons, 64.7%) significantly preferred curve and corner stimuli over single bars, and only 1.5% (eight neurons) significantly preferred bars over curves and corners (*Figure 3—figure supplement 1A*), indicating that neurons in these areas were indeed much more likely to encode more complex shape features compared to simple orientation. Therefore, we made a combined cell map to depict curve and corner selectivity (*Figure 3A*), neglecting bar responses, by calculating curve/corner index (CVCNI). Similar to CVSI and CNSI, positive CVCNI values indicate a neuron's maximum response to curves is stronger than its maximum response to corners, and vice versa. As expected, neurons with similar CVCNI values were spatially clustered. Neurons that fell into the 4×-defined curve domains generally had positive CVCNI values (*Figure 3B*) and those in the 4× corner domains generally had negative CVCNI values (*Figure 3C*). We also performed a one-way ANOVA comparing neurons' maximum curve and corner responses. We found that neurons with CVCNI > 0.2 or < −0.2 (which means 1.5 times as strong) predominantly showed significant preferences ($p < 0.05$) to curves or corners over the other kind (*Figure 3D*). The curve- or corner-selective neurons (red and blue neurons in *Figure 3D*) have very diverse curve or corner tuning, and could be either selective or invariant to the radius and radian of curves or bar length and separation angle of corners (*Figure 3—figure supplement 1B–D*), which potentially enables the encoding of multiple shape segments. More interestingly, these neurons that were heavily biased to curves or corners over the other tended to respond very weakly to single bars (*Figure 3E*), implying that they might be detecting more complex and integral shape features instead of local orientation. These results suggest that curves and corners are encoded by different neuronal clusters organized in curve and corner domains, and these domains are distinct from those representing single orientations.

## Curve-preferring neurons are selective for smoothness

Curves and corners are both different from single bars in that they potentially contain multiple different local orientations, yet we found them to be encoded by different neuronal clusters in V4. This suggests that V4 neurons are not recognizing shapes with more than one local orientation, but computing a more fundamental feature difference. To investigate what distinguishes curves from corners in V4, we tested hexagonal segments (Π-shape stimuli; *Figure 4A*) that highly resemble curves except for a lack of smoothness (*Nandy et al., 2013*). We found that neurons that were very selective to smooth curves did not respond strongly to Π-shape stimuli (*Figure 4A*), suggesting that they were selective to smoothness, rather than multiple orientations. In the same way as CVCNI, we calculated curve/Π-shape index (CVPII), which characterizes a neuron's preference to smooth curves over the Π-shape stimuli. We found that neurons' CVPII were highly consistent with CVCNI (R = 0.72, $p < 0.001$, *Figure 4B*), which means neurons preferring smooth curves over corners would also prefer smooth curves over Π-shape stimuli. As a result, the maps of CVPII were also consistent to CVCNI maps (*Figure 4C* vs. *Figure 3A*). K-means clustering analysis of population responses also showed that smooth curves are encoded differently from rectilinear shapes including Π-shapes and corners (*Figure 4—figure supplement 1*). Therefore, smoothness is important to the distinct encoding of curves and corners in the specific curve domains and corner domains in V4.

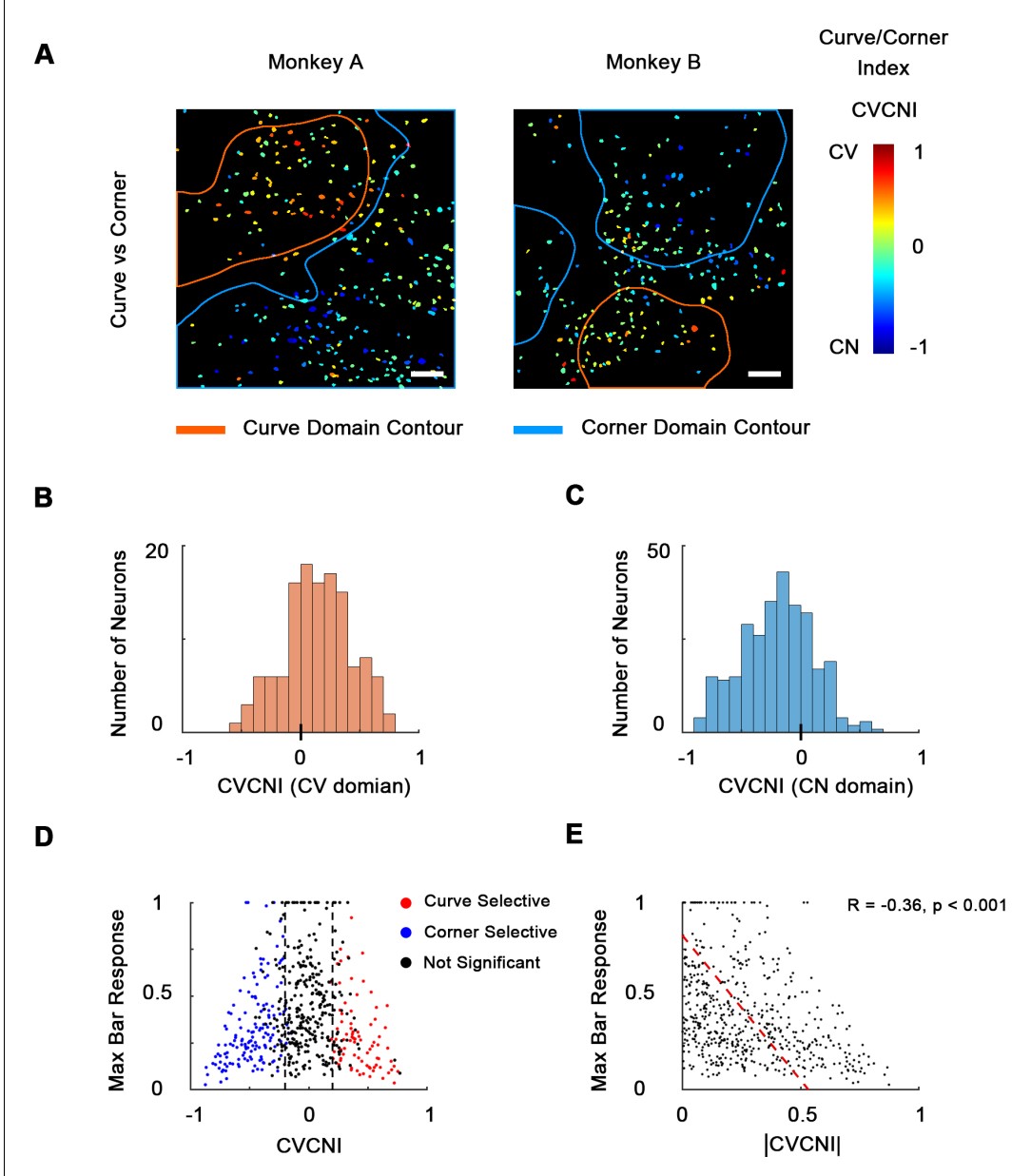

**Figure 3.** Combined maps of curve/corner preference. (**A**) Cell map of curve/corner index (CVCNI). Positive CVCNI indicates preference for curves over corners and vice versa. Curve-selective neurons and corner-selective neurons are spatially clustered. Scale bar = 100 μm. (**B**) Histogram of CVCNI for neurons located within the curve-biased domains. Mean = 0.15 ± 0.03 S.E. (**C**) Histogram of CVCNI for neurons located within the corner-biased domains. Mean = −0.20 ± 0.02 S.E. (**D**) Scatterplot of maximum responses to bars (normalized to 0–1 by the maximum responses to all contour features) against CVCNI. Red dots indicate neurons showing significant preference for curves (ANOVA p<0.05, n = 10) and blue for corners. The majority of neurons (74.5%) with CVCNI < −0.2 or >0.2 were significantly selective. Neurons that highly preferred curves over corners or corners over curves did not respond strongly to single-orientated bars. (**E**) Neurons' maximum bar responses were negatively correlated with the absolute values of CVCNI. The red line represents the linear regression line.

The online version of this article includes the following figure supplement(s) for figure 3:

**Figure supplement 1.** Neurons' tuning to curves and corners.

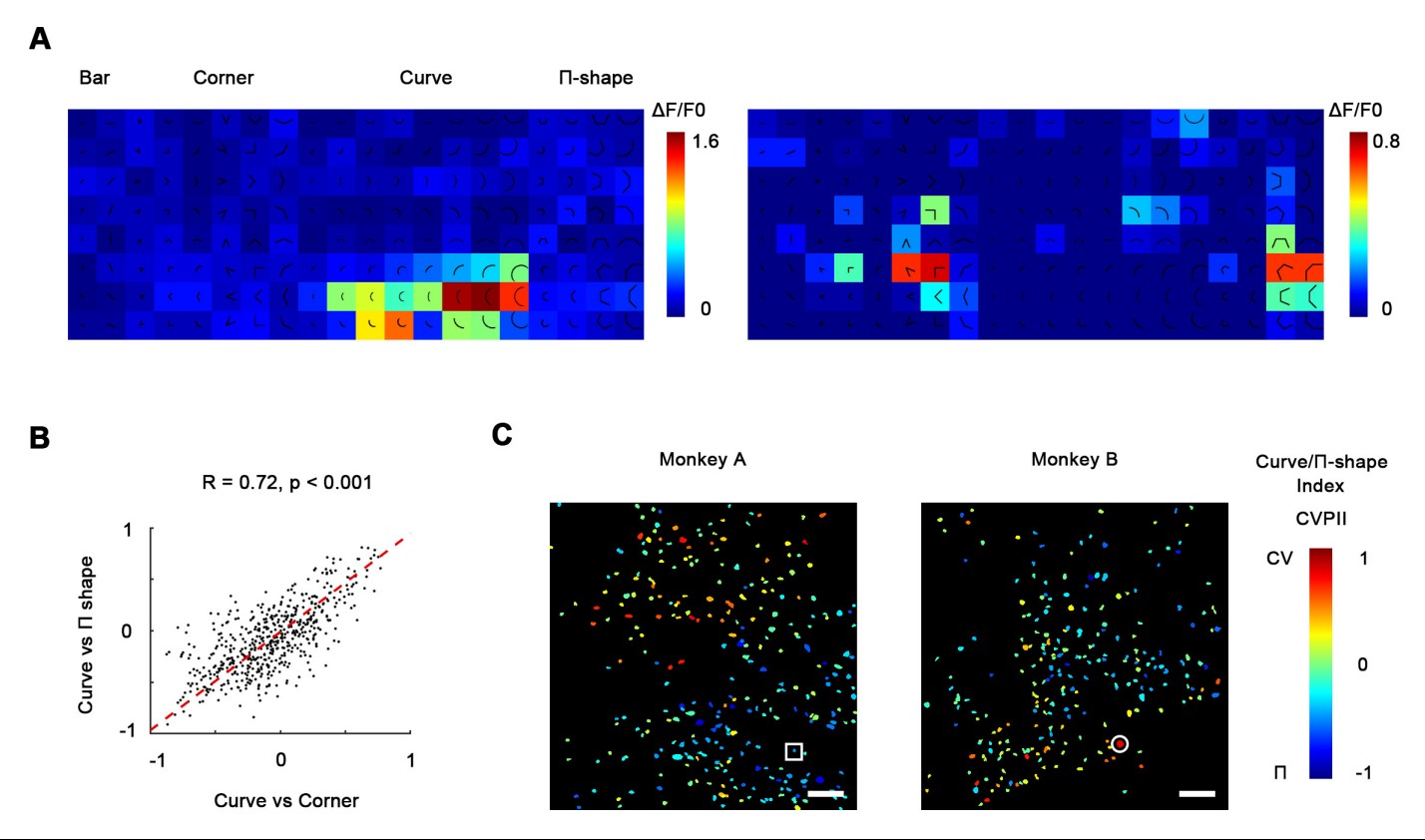

**Figure 4.** Curve-preferring neurons are selective for smoothness. (A) Left: responses of an example curve preferring neurons to bars, corners, smooth curves, and Π-shape stimuli, indicated by the white circle in (C). The neurons responded strongly to smooth curves but not to Π-shape, which highly resemble curves despite lack of smoothness. Right: an example neuron responding to rectilinear corners and Π-shapes, indicated by the white square in (C). (B) Scatterplot of curve/corner index (CVCNI) against curve/Π-shape index (CVPII), which characterizes neuronal preference for smooth curves over Π-shape stimuli. The red dash line represents the linear regression line. The two values were highly correlated, indicating that neurons preferring curves over corners also preferred curves over Π-shape stimuli. (C) Cell map of CVPII. Scale bar = 100 μm. Neurons are clustered similarly to CVCNI (*Figure 3A*).

The online version of this article includes the following figure supplement(s) for figure 4:

**Figure supplement 1.** K-means clustering analysis.

## Curve and corner selectivity is related to concentric and radial grating preference

Early studies in V4 demonstrated that many V4 neurons are selective for non-Cartesian gratings (*David et al., 2006*; *Gallant et al., 1993*; *Gallant et al., 1996*). While concentric gratings highly resemble curves and radial gratings resemble corners, this result highly implied the curve/corner preference. Therefore, we wondered whether these two types of gratings are also separately encoded by neurons in curve and corner domains. So in addition to contour feature stimuli, we also tested concentric, radial, and Cartesian gratings (*Figure 5—figure supplement 1A*). The resultant selectivity maps were consistent with the contour feature maps as predicted. 48.4% of the neurons recorded in the imaging areas significantly preferred concentric or radial gratings over Cartesian gratings, while only 2.2% significantly preferred Cartesian gratings (*Figure 5—figure supplement 1B*). In addition, many of them were heavily biased to one over the other. Similar to CVCNI, we computed concentric/radial index (CRI) to characterize this bias. CRI and CVCNI values were found to be correlated (R = 0.38, p<0.001; *Figure 5B, Figure 5—figure supplement 2*), and naturally their cell maps were also consistent (*Figure 5C* vs. *Figure 3A*), suggesting that classical polar grating selectivity is closely related to curve and corner selectivity. Meanwhile, to assess whether the observed selectivity is related to different spatial frequencies, we examined the CRI map at 1, 2, and 4 cycle/°.

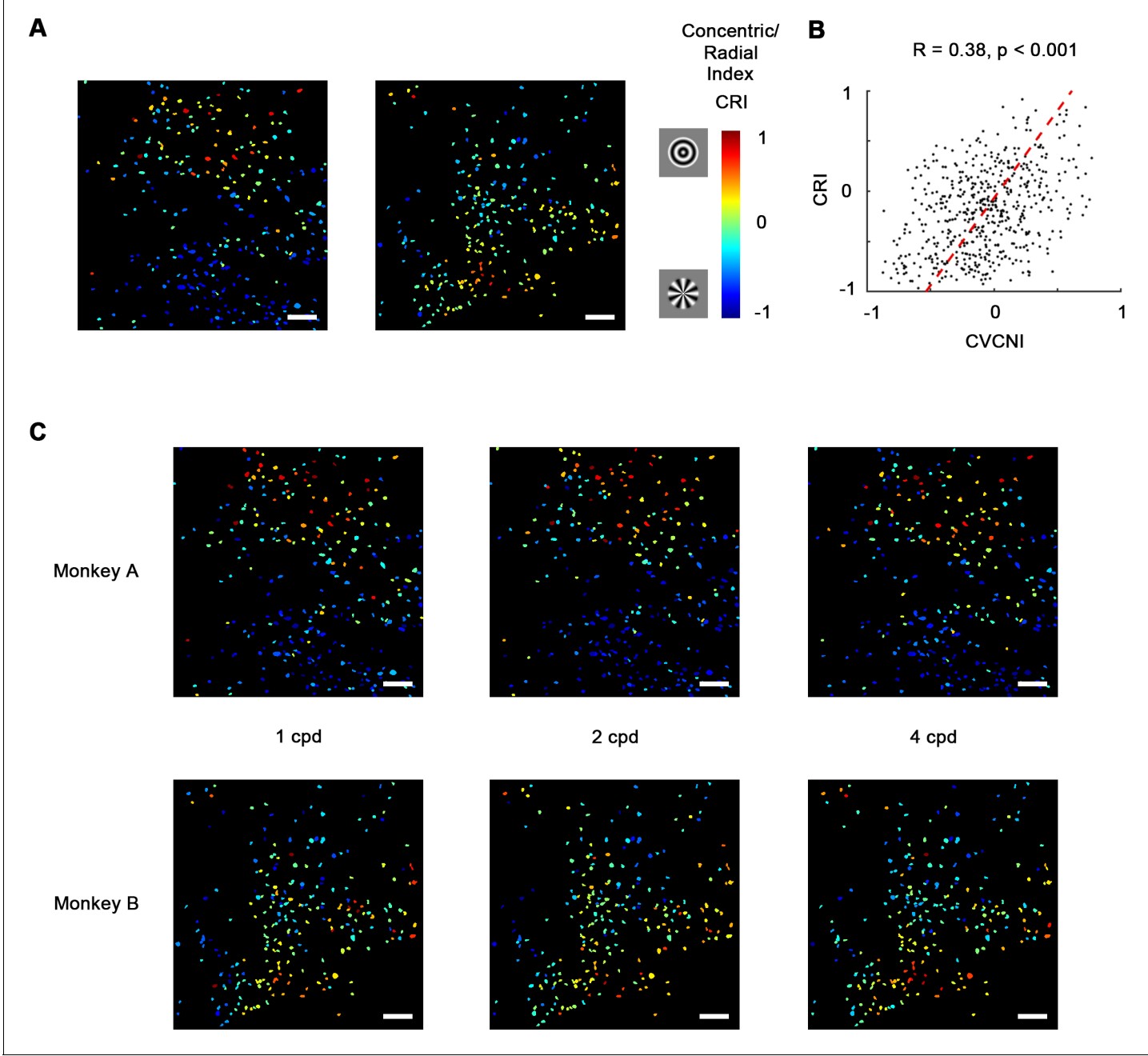

**Figure 5.** Concentric and radial gratings preference. (**A**) Cell map of concentric/radial index (CRI). Positive CRI indicates preference for concentric over radial gratings and vice versa. Concentric grating-selective neurons and radial grating-selective neurons are spatially clustered, and the overall distribution was consistent with curve/corner selectivity (*Figure 3A*). Scale bar = 100 μm. (**B**) Scatterplot of curve/corner index (CVCNI) against CRI, which were positively correlated. The red dash line represents the linear regression line. (**C**) CRI cell maps at spatial frequencies of 1, 2, and 4 cycles/° (cpd). The map structure remained consistent.

The online version of this article includes the following figure supplement(s) for figure 5:

**Figure supplement 1.** Responses to Cartesian, concentric, and radial gratings.

**Figure supplement 2.** Concentric/radial index (CRI).

The CRI values of all neurons at three spatial frequencies are highly correlated (Pearson correlation, all R > 0.5, p<0.001), and the map structures were found to remain consistent across three spatial frequencies (*Figure 5C*), implying such selectivity is not directly related to spatial frequency.

## Discussion

Using 2P calcium imaging, we identified cortical patches in macaque V4d selective for curves or corners (*Figure 1E, F*), with individual curve- and corner-selective neurons consistently clustered spatially (*Figure 3A*). These neurons exhibited diverse curve or corner selectivity (*Figure 3—figure supplement 1B–D*) and could potentially be involved in the encoding and processing of a large variety of curves and corners. These results demonstrate the existence of functionally specific curve and corner domains in V4d.

Functional organization for low-order orientation and spatial frequency representations in macaque V4 had previously been visualized using ISOI (*Lu et al., 2018*; *Roe et al., 2012*). For more complex shape features, very few studies have been carried out in V4 to characterize its functional organization, let alone at single-cell resolution. We report here the existence of cortical micro-domains consisting almost entirely of neurons selective for curves. This finding at the single-cell level is consistent with an fMRI study that reported curvature-biased patches in macaque V4 (*Yue et al., 2014*). The patches we found were smaller in size (about 300 μm) than those observed using fMRI; we suspect due to the improved spatial resolution afforded by 2P imaging. Additionally, we also found cortical domains in V4d selective for corners. Apart from this fMRI study, one of the reasons why exactly the curve/corner contrast was used to study functional domains in V4 was that in our recent study using natural images we found smooth curves and rectilinear corners to be one of the dominant features encoded by many V4 neurons (*Jiang et al., 2019*). Here, we directly demonstrated and visualized the combined functional organization of smooth curves and rectilinear corners in V4 at both cortical and single-cell level.

Two recent papers have also reported curvature domains in anesthetized macaque V4. *Hu et al., 2020* used ISOI, finding functional domains that prefer curved over straight gratings. *Tang et al., 2020* used both ISOI and 2P imaging, finding functional domains that prefer circles over rectilinear triangles. In general, these two imaging studies alongside our own provide clear replication of the core importance of curvature as an organizing principle in the functional architecture of V4. Compared to ISOI, 2P imaging holds the advantage of higher spatial resolution, and therefore makes it possible to characterize the transition between domains more precisely than Gaussian smoothed ISOI. We found the transition taking place within around 300 μm, remaining relatively elevated thereafter (*Figure 2—figure supplement 3*, *Figure 5—figure supplement 2*). Comparing the different stimulus set (curves vs. corners, concentric vs. radial), the transition of CRI maps of monkey A in *Figure 5A* looked sharper probably because too many neurons had negative CRI values. CRI and CVCNI were correlated but not identical. Since concentric gratings only have 360° full circles but some neurons might prefer short arches (small radian, *Figure 3—figure supplement 1*), it is possible that they do not respond strongly to concentric gratings and tend to have negative CRI.

A number of electrophysiology studies have reported that some neurons in V4d are selective for more complex features (*Gallant et al., 1993*; *Hegdé and Van Essen, 2007*; *Kobatake and Tanaka, 1994*; *Pasupathy and Connor, 1999*). Our results, consistent with these works, identified many curve- or corner-selective neurons. In addition, given the ability of 2P imaging to quantify the spatial relationships between neurons, we confirm that they are spatially clustered. We also observed some deviations of our results from earlier studies. First, the percentage of complex feature-selective neurons we found in our study is higher than previously observed (*Gallant et al., 1993*; *Pasupathy and Connor, 1999*); in our hands, the vast majority of neurons preferred curves and corners over bars and concentric and radial gratings over Cartesian gratings. Second, although Π-shape stimuli were sometimes regarded also as curved contours (*Nandy et al., 2013*), we found V4 neurons responding to them differently. We think these two deviations are primarily due to sampling neurons within or close to curve and corner domains (which is difficult to detect with classical electrophysiology). We do not wish to infer that curve and corner stimuli are only encoded by neurons in the curve and corner domains while other neurons are not involved. But we have demonstrated that neurons in the curve and corner domains are tuned to more complex and integral features rather than local orientation, spatial frequency, or multiple orientations, alone, supporting the encoding of shape segments

with intermediate complexity in V4 (*Bushnell and Pasupathy, 2012*; *El-Shamayleh and Pasupathy, 2016*; *Oleskiw et al., 2014*; *Rust and DiCarlo, 2010*).

Complexity increases as visual shape information is processed along the ventral visual pathway. Neurons in V1 are tuned to low-order orientation and spatial frequency and organized in iso-orientation domains and orientation pinwheels (*Nauhaus et al., 2012*; *Ts'o et al., 1990*). Neurons in IT are selective for complex features and objects and organized in feature columns and face patches (*Tanaka, 2003*; *Tsao et al., 2006*; *Tsunoda et al., 2001*). The simple-to-complex transformation and integration take place in the intermediate stages between V1 and IT. Researchers have reported that some V2 neurons are selective to combination of multiple local orientations, from which corner selectivity might emerge (*Anzai et al., 2007*; *Ito and Komatsu, 2004*). Our results in V4d showed that intermediate shape segments like curves and corners are separately encoded by neurons in specific functional domains, and the curve- and corner-selective neurons are tuned to the integral features instead of local orientation or combination of orientations. It is possible that these complex feature-selective neurons receive inputs or modulation from nearby neurons or downstream areas to form a recurrent network, which might underlie previous findings that the response profiles of V4 neurons were temporally heterogeneous (*Nandy et al., 2016*; *Yau et al., 2013*). Such evidence is also recently accumulating for IT cortex (*Kar and DiCarlo, 2021*). Unfortunately, this question is difficult to address given the temporal resolution of the existing calcium imaging technique. One possible solution is to use genetically encoded voltage indicators (*Xu et al., 2017*; *Yang and St-Pierre, 2016*), which once successfully applied in macaques could help to reveal the simple-to-complex integration of neurons.

Given that we recorded neurons whose stimulation was not isolated to the 'optimal' spatial location in the receptive fields (i.e., the RF locations of some neurons might deviate for the population RF), the nature of the domains may also be modulated by stimulus translation variance, and future studies addressing positional variance and stimulus encoding are warranted. Our sample of V4d was also near-foveal in terms of eccentricity. It is well established that the ventral pathway connectivity to IT favors central rather than peripheral visual space (*Ungerleider et al., 2008*), but the relationship of visual eccentricity to these functional domains remains unknown. The existence of curve and corner domains for neuronal encoding in V4d provides significant support for integration of shape information in the intermediate stages of the visual hierarchy. These findings provide a more comprehensive understanding of the functional architecture of V4 feature selectivity.

Finally, our results may also help to explore the later stage of the visual hierarchy. The data suggests that higher-order pattern domains may emerge gradually along the ventral pathway. The specificity of clustered patches/domains in the cortex has been proposed as an important organizing principle for some, though not all, domains of cortical processing (*Kanwisher, 2010*). A recent study has suggested that, at least for faces and color processing, such functional domains are causally specific for human visual recognition (*Schalk et al., 2017*). The curve and corner domain responses in V4 could possibly form the basis for more complex feature columns, object domains, and face patches in IT. This is consistent with a growing body of evidence from the ventral stream (*Bao et al., 2020*; *Rajalingham and DiCarlo, 2019*; *Yue et al., 2014*). Recent explorations of neuronal response fields in artificial neural networks have likewise found a prevalence of curve detectors with increasing complexity along the processing hierarchy (*Cammarata et al., 2020*). Studying such functional cross-areal connectivity (both bottom-up and top-down) remains a critical goal for future studies of the visual system. It is also interesting to try to identify why smooth curves and rectilinear corners are separated as early as V4. One possible explanation is that smooth curves are more prevalent in living animals or foods that are of particular interest to primates, while corners are often found in the background environment of stones or branches. Such differences may underlie the statistical regularities in natural images of objects (*Long et al., 2018*; *Levin et al., 2001*; *Yetter et al., 2020*; *Zachariou et al., 2018*). Such comparisons will provide a basis for future investigations comparing the statistical feature relationships for natural images between V4 and IT functional domains.

## Materials and methods

**Key resources table**

*Continued*

| Reagent type (species) or resource | Designation | Source or reference | Identifiers | Additional information |
|---|---|---|---|---|
| Reagent type (species) or resource | Designation | Source or reference | Identifiers | Additional information |
| Strain, strain background (*Macaca mulatta*) | *Macaca mulatta* | Beijing Prima Biotech Inc | | http://www.primasbio.com/en/Home |
| Recombinant DNA reagent | AAV9.Syn. GCaMP6f.WPRE. SV40 | Penn Vector Core | CS1001 | |
| Recombinant DNA reagent | AAV1.Syn. GCaMP5G. WPRE.SV40 | Penn Vector Core | V4102MI-R | |
| Software, algorithm | MATLAB R2018b | MathWorks | | https://www.mathworks.com |
| Software, algorithm | Code for data analysis | This paper | | https://github.com/RJiang1994/macaque-v4-2P (*Jiang, 2021* copy archived at swh:1:rev:57dfeac5e81b91c93ef0687f8cf04010d3f47f8c) |

All procedures involving animals were in accordance with the Guide of Institutional Animal Care and Use Committee (IACUC) of Peking University Laboratory Animal Center and approved by the Peking University Animal Care and Use Committee (LSC-TangSM-5).

## Animal preparation

The subjects used in this study were two adult male rhesus monkeys (*Macaca mulatta*, 4 and 5 years of age, respectively), purchased from Beijing Prima Biotech Inc and housed at Peking University Laboratory Animal Center. Two sequential surgeries were performed on each animal under general anesthesia. In the first surgery, we performed a craniotomy over V4 and opened the dura. We injected 200 nl of AAV9.Syn.GCaMP6f.WPRE.SV40 (CS1001, titer 7.748e13 [GC/ml], Penn Vector Core) or AAV1.Syn.GCaMP5G.WPRE.SV40 (V4102MI-R, titer 2.37e13 [GC/ml], Penn Vector Core) at a depth of about 350 µm and speed of 5–10 nl/s. Injection and surgical protocols followed our previous study (*Li et al., 2017*). After injections, we sutured the dura, replaced the skull cap with titanium screws, and closed the scalp. The animal was then returned for recovery and received Ceftriaxone sodium antibiotic (Youcare Pharmaceutical Group Co. Ltd., China) for 1 week. 45 days later, we performed the second surgery to implant the imaging window and head posts. The dura was removed and a glass coverslip was put directly above the cortex without any artificial dura and glued to a titanium ring. We then glued the titanium ring to the skull using dental acrylic. The detailed design of the chamber and head posts can be found in our previous study (*Li et al., 2017*). Monkeys can be ready for recording about 1 week after the second surgery.

## Behavioral task

Monkeys were trained to maintain fixation on a small white spot (0.1°) while seated in a primate chair with head restraint to obtain a juice reward. Eye positions were monitored by an ISCAN ETL-200 infrared eye-tracking system (ISCAN Inc, Woburn, MA) at a 120 Hz sampling rate. Trials in which the eye position deviated 1° or more from the fixation point were terminated and the same condition was repeated immediately. Only data from the successful trials was used.

## Visual stimuli

The visual stimuli were displayed on an LCD monitor 45 cm from the animal's eyes (Acer v173Db, 17 inch, 1280 × 960 pixel, 30 pixel/°, 80 Hz refresh rate). After acquiring fixation, only the gray

background (32 cd/m$^2$) was presented for the first 1 s to obtain the fluorescence baseline, and then the visual stimuli were displayed for further 1 s. No inter-trial interval was used. Stimuli were presented in pseudo-random order. We used square-wave drifting gratings (0.4° diameter circular patch, full contrast, 4 cycle/°, 3 cycle/s) generated and presented by the ViSaGe system (Cambridge Research Systems, Rochester, UK) to measure the retinal eccentricity, which was about 1° bottom left to the fovea for both monkeys.

Contour feature stimuli were generated using MATLAB (The MathWorks, Natick, MA) and presented using the ViSaGe system (Cambridge Research Systems). The contour feature stimuli were two pixels wide. The lengths of the bars and corner edges were 10 and 20 pixels (30 pixel/°, 0.33° and 0.67°), and the radius of curve stimuli were also 10 and 20 pixels. For each of the two sizes, the curve stimuli varied in radians (120°, 180° for 4× imaging and 60°, 90°, 120°, 180° for 16× imaging). The corner stimuli also varied in three separation angles (45° and 90° and 135°). All contour feature stimuli were rotated to eight orientations (0°, 45°, 90°, 135°, 180°, 225°, 270°, 315° for curves and corners, and 0°, 22.5°, 45°, 67.5°, 90°, 112.5°, 135°, 157.5° for bars).

The Cartesian (eight orientations, 0°, 45°, 90°, 135°, 180°, 225°, 270°, 315°), concentric, and radial grating stimuli were full contrast sinusoidal gratings (edge blurred), which were 90 pixels (3°) in diameter, with spatial frequencies (SF) of 1, 2, and 4 cycle/°. The concentric gratings were generated as

$$CG = sin\left(2\pi SF * \sqrt{x^2 + y^2}\right)$$

The radial gratings were generated as

$$RG = sin\left(2\pi SF * \arctan\left(\frac{y}{x}\right)\right)$$

The data for contour feature stimuli was recorded on one day, and the data for gratings on another day.

## 2P imaging

2P imaging was performed using a Prairie Ultima IV 2P laser scanning microscope (Bruker Corporation, Billerica, MA) during experiments. 1000 nm mode-lock laser (Spectra-Physics, Santa Clara, CA) was used for excitation of GCaMPs, and resonant galvo scanning (512 × 512 pixel, 32 frame/s) was used to record the fluorescence images (8 fps, averaging every four frames). A 4× objective (Nikon Corporation, Tokyo, Japan) was used for sub-cortical-level recording (3.4 × 3.4 mm, 6.7 µm/pixel), and a 16× objective (Nikon Corporation) for neural population recording at single-cellular resolution (850 × 850 µm, 1.7 µm/pixel). We used a Neural Signal Processor (Cerebus system, Blackrock Microsystem, Salt Lake City, UT) to record the time stamp of each frame of the 2P images as well as the time stamps of visual stimuli onset for synchronization.

## Image data processing

Image data was processed by MATLAB. The 2P images were first aligned to a template image by a 2D cross-correlation algorithm (*Li et al., 2017*) to eliminate motion artifacts during recording sections. For all the successful trials, we found the corresponding 2P images by synchronizing the time stamps of stimulus onset recorded by the Neural Signal Processor (Cerebus system, Blackrock Microsystem). The differential fluorescence image was calculated as ΔF = F – F0, where the basal fluorescence image F0 was defined as the average image of 0–0.5 s before stimulus onset, and F as the average of 0.5–1.25 s after stimulus onset, both averaged across all repeats for each stimulus.

For 4× imaging, the ΔF/F0 maps were Gaussian smoothed using a low-pass Gaussian filter (σ = 10 pixels) to obtain the activation maps. For 16× imaging, to identify responding cell bodies (ROIs), the differential image (ΔF) for each stimuli went through a band-pass Gaussian filter (σ = 2 pixels and 5 pixels, respectively, only used for identifying ROIs) and were then binarized using a pixel value threshold of 3 SD. The connected components (>25 pixels) were identified as candidates for active ROIs. An ROI was discarded if its maximum response (ΔF/F0) was below 0.3. The roundness of these ROIs was calculated as

$$C = \frac{P^2}{4\pi S}$$

where P is the perimeter of the ROI and S is the area. Only ROIs with C < 1.1 were identified as cell bodies. We also tested this criterion by ANOVA, comparing the fluorescence 0–0.5 s before and 0.5–1.25 s after stimulus onset (same definition as ΔF), all trials together. 533 out of the 535 neurons identified had p<0.05.

## Curve and corner domains

All trials (stim number × repeat number) in 4× imaging were categorized first as curves (32 stim), corners (48 stim), or bars (16 stim). Curve patches: for each pixel, independent t-tests were performed to compare the responses to all curves against all corners and against all bars, respectively, and the larger one of the two p-values was chosen if the mean response to curves is stronger than corners and bars. FDR was computed following a Benjamini–Hochberg procedure, using the MATLAB command mafdr, in which $q_i = p_i \times 512 \times 512 / \text{rank}(p_i)$. Corner patches followed the same procedure.

Cluster permutation tests were then performed to exclude patches with too few significant pixels. For each permutation, all trials (stim number × repeat number) were randomly relabeled as curves, corners, or bars, keeping the total trial number within each of the three groups unchanged. Independent t-tests as in *Figure 1F* were performed, with an uncorrected p=0.01 as threshold. The cluster (connected component) with the maximum pixel number was recorded. 60,000 random permutations were performed, resulting in 60,000 maximum cluster sizes as null distribution. The top 5% (3000) of the null distribution was used as the threshold, and the patches with pixels below this level were regarded as insignificant and excluded.

## Quantification and statistical analysis

Two tests were performed to determine whether a neuron was selective to the orientation of curves or corners. First, we performed ANOVA to compare the fluorescence 0–0.5 s before and 0.5–1.25 s after stimulus onset (same definition as ΔF) using all the trials for curve and corner stimuli. Then we find the optimal curve or corner stimuli of this neuron and used ANOVA to compare among the eight orientations of this optimal form. The p-value was then Bonferroni-corrected (14 comparisons, 6 corners, and 8 curves). Only neurons passing both ANOVA tests (p<0.05) were deemed as tuned to the orientation of curves or corners.

CVSI is used to characterize a neuron's preference to curves over other stimuli (bars and corners), defined as

$$\text{CVSI} = \frac{MaxResp_{\text{curve}} - MaxResp_{\text{other}}}{MaxResp_{\text{curve}} + MaxResp_{\text{other}}}.$$

where MaxResp$_{\text{curve}}$ is the neuron's maximum response to curve stimuli and **MaxResp**$_{\text{other}}$ is the neuron's maximum response to other stimuli (bars and corners). CVSI ranges from −1 to 1, and a positive CVSI value indicates a neuron's response to its optimal curve stimuli is greater than its response to optimal bar or corner stimuli.

CNSI is defined as

$$\text{CNSI} = \frac{MaxResp_{\text{corner}} - MaxResp_{\text{other}}}{MaxResp_{\text{corner}} + MaxResp_{\text{other}}}.$$

where MaxResp $_{\text{corner}}$ is the neuron's maximum response to corner stimuli and **MaxResp**$_{\text{other}}$ is the neuron's maximum response to other stimuli (bars and curves).

CVCNI is defined as

$$\text{CVCNI} = \frac{MaxResp_{\text{curve}} - MaxResp_{\text{corner}}}{MaxResp_{\text{curve}} + MaxResp_{\text{corner}}}.$$

We also performed one-way ANOVA test comparing neuron's maximum response to curve stimuli and maximum response to corner stimuli in *Figure 3D*, with threshold value p=0.05, repeats n = 10. The same tests were also applied to CVSI and CNSI in *Figure 2—figure supplement 3*.

CVPII is defined as

$$\mathrm{CVPII} = \frac{MaxResp_{\mathrm{curve}} - MaxResp_{\Pi-\mathrm{shape}}}{MaxResp_{\mathrm{curve}} + MaxResp_{\Pi-\mathrm{shape}}}.$$

where MaxResp$_{\Pi\text{-shape}}$ is the neuron's maximum response to $\Pi$-shape stimuli. The Pearson correlation of CVCNI and CVPII was calculated in **Figure 4B**, and the regression line was derived by minimizing $\sum \sqrt{(\Delta\mathrm{x})2 + (\Delta\mathrm{y})2}$.

CRI is defined as

$$\mathrm{CRI} = \frac{MaxResp_{\mathrm{concentric}} - MaxResp_{\mathrm{radial}}}{MaxResp_{\mathrm{concentric}} + MaxResp_{\mathrm{radial}}}.$$

where MaxResp$_{\mathrm{concentric}}$ is the neuron's maximum response to concentric gratings and **MaxResp$_{\mathrm{radial}}$** is the neuron's maximum response to radial gratings. The Pearson correlation of CVCNI and CRI was also calculated in **Figure 5B**, and the regression line was derived by minimizing $\sum \sqrt{(\Delta\mathrm{x})2 + (\Delta\mathrm{y})2}$.

## Clustering analysis

We analyzed 2922 neuron pairs from monkey A and 2432 neuron pairs from monkey B in **Figure 2I–K**. Pairwise tuning correlation was calculated as the Pearson correlation of the two neurons' responses to all bar, curve, and corner stimuli, and were plotted against pairwise spatial distances (averaging all neurons every 100 µm).

Similarly, the differences in CVSI and CNSI were also plotted against pairwise spatial distances:

$$\left|\Delta\mathrm{CVSI}_{ij}\right| = \left|\mathrm{CVSI}_i - \mathrm{CVSI}_j\right|,$$

where CVSI$_i$ is the CVSI of neuron $i$ and CVSI$_j$ is the CVSI of neuron $j$.

$$\left|\Delta\mathrm{CNSI}_{ij}\right| = \left|\mathrm{CNSI}_i - \mathrm{CNSI}_j\right|,$$

where CNSI$_i$ is the CVSI of neuron $i$ and CNSI$_j$ is the CVSI of neuron $j$.

Permutation test was performed to evaluate the significance of each average |ΔCVSI| and |ΔCNSI|. |ΔCVSI| or |ΔCNSI| were randomly paired with distances for 100,000 times to build the null distribution and averaged. A point was considered significant if it is higher than the top 100 of the null distribution or lower than the bottom 100 (p<0.001).

## K-means analysis

We performed K-means analysis to cluster the stimulus forms and the neurons. Responses of 535 neurons to 20 forms (two bars, eight curves, six corners, and four $\Pi$-shapes, each at eight orientations) are used to construct the responses matrix R as

$$\mathrm{R} = \begin{bmatrix} r_{1,1} & \cdots & r_{1,535} \\ \vdots & \ddots & \vdots \\ r_{20,1} & \cdots & r_{20,535} \end{bmatrix}$$

where $r_{i,j}$ is the response of neuron $j$ to stimulus form $i$. Only the maximum responses among eight orientations were used.

We used population response vectors (*RP*, rows of matrix R) to cluster the forms. For form $i$,

$$RP_i = \left( r_{i,1} r_{i,2} \cdots r_{i,535} \right).$$

We used neuron response vectors (*RN*, columns of matrix R) to cluster the neurons. For neuron $j$,

$$RN_j = \left( r_{1,j} r_{2,j} \cdots r_{20,j} \right).$$

The number of clusters was determined using Calinski–Harabasz criterion and squared Euclidean distance. Maximum literation time = 10,000. Clustering was repeated for 10,000 times with new initial cluster centroid, and the one with the lowest within-cluster sum was used.

## Multi-dimensional scaling

Classical multi-dimensional scaling (MDS) was performed to visualize the clustering of stimulus forms derived by K-means. The distance (dissimilarity matrix) was computed as

$$D_{i,j} = 1 - corrcoef\left(RP_i, RP_j\right)$$

where $D_{i,j}$ is the distance between form $i$ and $j$, $corrcoef$ is the Pearson correlation, and $RP_i$ is the population response vectors of form $i$. Classical MDS was performed using singular value decomposition (SVD) algorithm.

The normalized stress was computed as

$$\text{Stress} = \frac{\sum\left(D_{i,j} - D'_{i,j}\right)^2}{\sum D_{i,j}^2}$$

where $D_{i,j}$ is the distance in the original space and $D'_{i,j}$ is the distance in the new MDS space.

# Acknowledgements

We thank the Peking University Laboratory Animal Center for animal care. We acknowledge the Genetically Encoded Calcium Indicator (GECI) project at Janelia Farm Research Campus Howard Hughes Medical Institute. We thank Niall Mcloughlin and Cong Yu for their comments and suggestion on the manuscript.

# Additional information

### Funding

| Funder | Grant reference number | Author |
|---|---|---|
| National Natural Science Foundation of China | 31730109 | Shiming Tang |
| National Basic Research Program of China | 2017YFA0105201 | Shiming Tang |
| National Natural Science Foundation of China | U1909205 | Shiming Tang |
| Beijing Municipal Commission of Science and Technology | Z181100001518002 | Shiming Tang |
| Peking-Tsinghua Center for Life Sciences | | Shiming Tang |

The funders had no role in study design, data collection and interpretation, or the decision to submit the work for publication.

### Author contributions

Rundong Jiang, Conceptualization, Data curation, Software, Formal analysis, Investigation, Writing - original draft; Ian Max Andolina, Formal analysis, Writing - review and editing; Ming Li, Investigation; Shiming Tang, Conceptualization, Resources, Data curation, Software, Supervision, Funding acquisition, Methodology, Project administration, Writing - review and editing

### Author ORCIDs

Rundong Jiang ![ID] https://orcid.org/0000-0002-9217-0749
Ian Max Andolina ![ID] http://orcid.org/0000-0001-9985-3414
Ming Li ![ID] http://orcid.org/0000-0001-5173-1602
Shiming Tang ![ID] https://orcid.org/0000-0003-0294-3259

## Ethics

Animal experimentation: All procedures involving animals were in accordance with the Guide of Institutional Animal Care and Use Committee (IACUC) of Peking University Laboratory Animal Center, and approved by the Peking University Animal Care and Use Committee (LSC-TangSM-5).

## Decision letter and Author response

Decision letter https://doi.org/10.7554/eLife.63798.sa1
Author response https://doi.org/10.7554/eLife.63798.sa2

# Additional files

## Supplementary files

• Transparent reporting form

## Data availability

The data and MATLAB codes used in this study can be found in GitHub (https://github.com/RJiang1994/macaque-v4-2P; copy archived at https://archive.softwareheritage.org/swh:1:rev:57dfeac5e81b91c93ef0687f8cf04010d3f47f8c).

The following dataset was generated:

| Author(s) | Year | Dataset title | Dataset URL | Database and Identifier |
|---|---|---|---|---|
| Jiang R, Tang S | 2020 | macaque-v4-2P | https://github.com/RJiang1994/macaque-v4-2P | GitHub, github.com/RJiang1994/macaque-v4-2P |

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
