## [Decision Letter]

**Acceptance summary:**

Two-photon imaging in area V4 of awake monkeys was used to characterize the organization of tuning for distinct shape elements (curves, corners, and bars). The authors use a combination of wide field/low resolution imaging, to visualize large scale organization, with smaller field/high resolution imaging, to measure tuning and organization of individual neurons underlying the wide field results. At both scales, they establish that most V4 neurons are more responsive to curves and corners than to bars, and they establish anatomical segregation between neurons tuned for curves and neurons tuned for bars. These findings advance our understanding of the topographic organization of neuronal feature selectivity in area V4 of the macaque monkey.

**Decision letter after peer review:**

Thank you for submitting your article "Clustered Functional Domains for Curves and Corners in Cortical Area V4" for consideration by *eLife*. Your article has been reviewed by 4 peer reviewers, one of whom is a member of our Board of Reviewing Editors, and the evaluation has been overseen by Tirin Moore as the Senior Editor. The following individuals involved in review of your submission have agreed to reveal their identity: Timo van Kerkoerle (Reviewer #2); Ed Connor (Reviewer #3); Jack L Gallant (Reviewer #4).

The reviewers have discussed the reviews with one another and the Reviewing Editor has drafted this decision to help you prepare a revised submission.

Summary:

A prominent aspect of the visual cortex is the topographic organization in feature dimensions such as orientation, color, motion etc. The present study uses 2-photon imaging in area V4 of awake monkeys, which is a novel application of 2-photon, to characterize the organization of tuning for established shape elements (curves, corners, and bars). The authors use a combination of wide field/low resolution imaging, to visualize large scale organization, with smaller field/high resolution imaging, to measure tuning and organization of individual neurons underlying the wide field results. At both scales, they establish that most V4 neurons are more responsive to curves and corners than to bars, and they establish anatomical segregation between neurons tuned for curves and neurons tuned for bars.

Overall the reviewers made positive comments about this study especially noting the technological advance and the application of a new high-resolution imaging modality to the question of topographic organisation in area V4, although reviewers also commented that the present study is largely a replication of previous work.

Nonetheless, because of the technology used here, the reviewers assess that the work is of significant interest. The main comments of the reviewers pertained to the statistical analyses in this manuscript, which will require extensive revisions and data analyses.

Essential revisions:

1. Statistics

Major improvements will be required on the level of statistical and data analyses. In light of these concerns, we require the authors publish the data and software underlying the figures so that the statistical analyses become transparent and can be verified by the reviewers.

Reviewers commented that statistical analysis is almost completely lacking and is potentially wrong where it is provided. Complex results such as the ones presented by the authors need to be accompanied by appropriate spatial statistics. This will likely require substantial revision to the data analysis and the text. If necessary, the authors should consult a statistical/data science specialist for advice on how to perform the statistical analyses. It remains unclear whether the main claims will survive after appropriate analysis.

More specifically, it is unclear whether the ANOVA tests for significance of curvature- and corner-selective patches has been performed correctly. It appears that the authors identified curvature-selective patches by subtraction, and then performed the ANOVA on these patches. It is unclear whether this procedure is correct and may amount to double-dipping because regions are pre-selected before statistics are run. This kind of analysis can dramatically increase the Type 1 error rate and lead to false conclusions. Therefore, the significance values that are reported here are likely far more extreme than they would be otherwise. Many tutorials regarding how to do these sorts of tests correctly can be found in the neuroimaging literature, where this sort of problem has been extensively discussed in the literature and where it is standard to address it appropriately. The authors should consult one of those tutorials and implement a strictly correct (probably FDR-based) procedure. For instance, here is a possible starting point (https://www.ncbi.nlm.nih.gov/pmc/articles/PMC3221040/).

Reviewers commented that the statistical analysis to determine the significance of clustering also appear to be problematic. In fact, it is not clear from the details of the paper what has been precisely done. It appears that the CVSI and CNSI were not evaluated statistically, but ANOVA was used to evaluate tuning in some way, which remains unclear. It furthermore appears that spatial clustering was not assessed statistically at all. The lack of statistics and unclarity about statistics does not meet the prevailing standards of the field. Single neuron tuning needs to be assessed with the correct statistical tests, as does spatial clustering. Given these data and the pre-selection methods that were used to identify targets for the high-resolution analysis, this could be tricky. The authors should consult a statistical/data science specialist for advice on how do to these analyses correctly.

In general the results with the 16x objective likely suffer from double dipping, as they are preselected, therefore the statistical claims about large-scale topographic organization remain unconvincing.

2. Limitations were noted about the fine-grained analyses with the 16x objective. A limitation of the present work is that there is only one pair of patches for each animal images at 16x. The analyses need to be also extended. The authors should further analyze the topographic organization at the local scale, is the transition sharp or gradual, what is the variability etc. It seems that there is a rather sharp boundary and that the tuning stays relatively flat, looking at the Figures 2A-D, 3A, 4C and seems particular clear in 5A and C (using concentric versus radial gratings). However, there is no real quantification of this. Figure 2I-K shows the tuning over distance, but this analysis seems to be performed without taking the shape of the domain into account. One possibility would be to show a similar plot, but where the axis is taken perpendicular to the boundary of the domain. It seems that the interpretation that the authors give of the data would predict that the selectivity shows a sharp transition at the boundary and stays elevated within the domain. Furthermore, it would be relevant to get an estimate of the averaged selectivity as well as the variability within the domain, separately for the two animals. Finally, it would be relevant to compare both the sharpness of the transition as well as the mean and variability with the domain between the different stimulus set (curves versus angles, and concentric versus radial gratings).

3. Data visualisation

The authors should show more raw data (high-resolution fluorescence images with the field of view used for the main analyses), as well as traces of fluorescence as a function of time as is standard with imaging to appreciate the quality of the fluorescence traces (over tens of seconds). In addition showing dF/F responses for single neurons to different stimuli would be important.

4. Bar tuning

It needs to be very clear if the small amount of bar tuning reported is only in the ROIs that are defined by subtracting bars (where this would be therefore expected) or overall, in the discussion it currently sounds like this is the case overall which was not clear from the results.

5. Choice of stimuli

The exact choice of stimulus needs to be discussed: why only black (other studies used only white stimuli), why only lines (not surfaces as in e.g. Pasupathy et al. study that is referred to), why no colors. Is it assumed that this will not matter for the results and why?

The bar length is matched to the radius of the curve stimuli, which implies to me that the overall number of black pixels is never matched for bars vs the other categories? The authors should discuss if this is a problem.

Do you expect more curve/corner functional domains if you use different color or luminance contrast, or do you expect the non-significantly curve/corner clustered parts of V4 to contain other functional domains?

6. Temporal dynamics

The imaging technique confines analyses to a late time window. If possible refer to literature demonstrating that response preferences remain similar across time for these stimuli, since tuning can be dynamic over time (e.g. Nandy et al. 2016, Issa and DiCarlo *eLife*).

7. Introduction and Discussion:

Intro and Discussion read quite well and a lot of the relevant literature is referred to. But Intro and Discussion could include further/more explicit clarification why exactly this contrast (curve/corner) was used to study functional domains (or is this just a starting point), what other functional domains there could be.

The paper needs to cite literature relating curve/corner to animate/inanimate contrasts you discuss (e.g. Zachariou et al. 2018, and other work from Yue lab). You may consider a brief discussion/mention the potential use or function of functional topographic clustering (e.g. Kanwisher, DiCarlo), which is proposed to be related to naturalistic experience that is also discussed here without references.

The history presented in the introductory section of this paper is very strange. The first paper that reported curvature tuning in V4 was the Gallant et al. 1993 paper that is cited ambiguously here. It is true that paper used gratings rather than curved lines, but a neuron that is selective for curved gratings is also likely selective for curved lines. A similar principle holds for the hyperbolic grating selectivity reported in Gallant et al. 1993. The authors should address this directly and acknowledge the relationship late in their paper.

Similarly, in their subsequent 1996 longer report Gallant et al. argued that neurons selective for curved and hyperbolic gratings were spatially clustered. The data presented in the paper under review is far better than the data that were available to Gallant et al. way back in 1996, but this result was anticipated by that earlier 1996 report, however this finding is not cited.

The authors should discuss the recent paper by Roe lab on curvature patches using intrinsic optical imaging has just been published in *eLife*: https://elifesciences.org/articles/57261. This paper is relevant for relevant the points above, as they claim that there is a smooth transition from rectilinear to low curvature to high curvature (figure 7).

The authors should furthermore discuss this recent *eLife* paper on curvature domains, using both intrinsic and 2-photon imaging: https://elifesciences.org/articles/57502

[Editors' note: further revisions were suggested prior to acceptance, as described below.]

Thank you for resubmitting your work entitled "Clustered Functional Domains for Curves and Corners in Cortical Area V4" for further consideration by *eLife*. Your revised article has been reviewed by 3 peer reviewers, and the evaluation has been overseen by Tirin Moore as the Senior Editor, and a Reviewing Editor.

The manuscript has been improved but there are some remaining issues that need to be addressed, as outlined below:

*Reviewer #2:*

The authors replied sufficiently to most of the comments.

One answer is not clear to me, in response to the comment:

"Some more details about the expression levels would be useful. Most importantly, it is unclear from Figure 1C-F how homogenous the expression was in the selected region. Could you show a separate image where it is possible to judge the level of expression? Also, would it be possible to give an estimate of the general expression levels in terms of percentage of total neurons, as well as the percentage of neurons that were nucleus filled? Finally, it would be relevant to know injection speed in this regard."

First of all, they still do not provide the injection speed.

Also, they write: "Most of the neurons that are clearly visible in an average image are not nucleus filled (Figure 1—figure supplement 2)."

However, Figure 1—figure supplement 2 does not show any individual cells. Nor do any of the other supplementary figures provide an image where it is possible to judge the structure of the labelling in individual cells, so allowing to see whether they have a clear donut shape, or are nucleus filled. It would therefore still be relevant to see a large / high resolution image where this can be judged.

*Reviewer #4:*

The authors have put a lot of work into this revision and the paper is substantially improved over the initial submission. The paper is still largely replicative and confirmatory, but there is a place in the literature for such papers.

It is reported that the V4 receptive fields sampled here were very close to the fovea. That implies that the viewing window was very far lateral, much farther than most prior V4 studies. My intuition is that the ear would have had to be removed in order to access V4 at this location. If the authors recorded more medially then I suggest that they recheck their reported eccentricity to be sure that it is correct.

The indexes that are used here have a pretty unintuitive and unusual scaling range. (For example, an index of 0.2 indicates a 1.5 times difference.) The paper would probably be easier to understand if they had a more intuitive range/form. (For example, if 1.5 indicated a 1.5 times difference.) However, this is up to the authors' discretion.

Figure 2I "significant" is misspelled. There are also a few places throughout the manuscript where pronouns are missing. (I commend the authors on the English though, it is generally quite good!)

Also in Figure 2, please spell out what "CVSI" and "CNSI" mean in the caption. In this and other captions, it is best if the reader can generally understand the caption on its own, w/o having to wade through the text.

The use of hexagonal segments to try to understand differences in tuning for curves versus angles is a weak approach, because hexagonal shapes are a poor intermediate model for these feature classes. A much more powerful method for understanding these differences would be to use an explicit computational model. But that seems to be beyond the scope of this paper…

---

## [Author Response]

Essential revisions:1. StatisticsMajor improvements will be required on the level of statistical and data analyses. In light of these concerns, we require the authors publish the data and software underlying the figures so that the statistical analyses become transparent and can be verified by the reviewers.Reviewers commented that statistical analysis is almost completely lacking and is potentially wrong where it is provided. Complex results such as the ones presented by the authors need to be accompanied by appropriate spatial statistics. This will likely require substantial revision to the data analysis and the text. If necessary, the authors should consult a statistical/data science specialist for advice on how to perform the statistical analyses. It remains unclear whether the main claims will survive after appropriate analysis.More specifically, it is unclear whether the ANOVA tests for significance of curvature- and corner-selective patches has been performed correctly. It appears that the authors identified curvature-selective patches by subtraction, and then performed the ANOVA on these patches. It is unclear whether this procedure is correct and may amount to double-dipping because regions are pre-selected before statistics are run. This kind of analysis can dramatically increase the Type 1 error rate and lead to false conclusions. Therefore, the significance values that are reported here are likely far more extreme than they would be otherwise. Many tutorials regarding how to do these sorts of tests correctly can be found in the neuroimaging literature, where this sort of problem has been extensively discussed in the literature and where it is standard to address it appropriately. The authors should consult one of those tutorials and implement a strictly correct (probably FDR-based) procedure. For instance, here is a possible starting point (https://www.ncbi.nlm.nih.gov/pmc/articles/PMC3221040/).

We agree that the map subtraction used in the previous version of the manuscript is more appropriate as a visualization of the data trends but not a strict statistical analysis. In the revised manuscript, we now have performed independent t-test to each pixel using Benjamini–Hochberg FDR correction and cluster permutation testing following the recommendations in Nichols and Holmes 2002 (Hum. Brain Mapp.). The new FDR q-value, replacing the previous SD maps, can be found in Figure 1F-G. The details of the new analysis can be found in the Methods – Curve and corner domains section line 491-507. In brief, p-value was computed for each pixel comparing the responses to all curves, all corners and all bars, and corrected by BH FDR (Figure 1 — figure supplement 3). Cluster permutation tests were performed to exclude patches (q-value<0.01) with too few pixels (Author response image 1). We used uncorrected p-values instead of q-values in permutation to build the null distribution and compared to the real FDR clusters because otherwise the cluster sizes in the permutations would be too small. This would result in larger cluster sizes in the null distribution and therefore larger critical value, and make the Type 1 error rate even lower.

As the shape of the domains are slightly changed using the corrected procedures, the example neuron 2 in ***Figure 2G*** of the old version is now outside the previous curve domain, so we have changed it to another neuron.

**Author response image 1. sa2fig1:** The null distribution of cluster permutation test (in descending rank order). The 3000^th^ (top 0.05) maximum cluster size (in pixels) is chosen as threshold.

Reviewers commented that the statistical analysis to determine the significance of clustering also appear to be problematic. In fact, it is not clear from the details of the paper what has been precisely done. It appears that the CVSI and CNSI were not evaluated statistically, but ANOVA was used to evaluate tuning in some way, which remains unclear. It furthermore appears that spatial clustering was not assessed statistically at all. The lack of statistics and unclarity about statistics does not meet the prevailing standards of the field. Single neuron tuning needs to be assessed with the correct statistical tests, as does spatial clustering. Given these data and the pre-selection methods that were used to identify targets for the high-resolution analysis, this could be tricky. The authors should consult a statistical/data science specialist for advice on how do to these analyses correctly.In general the results with the 16x objective likely suffer from double dipping, as they are preselected, therefore the statistical claims about large-scale topographic organization remain unconvincing.

We agree that CVSI and CNSI should be statistically analyzed. In fact, we only performed ANOVA to compare the responses to the optimal curve against the optimal corner (***Figure 3D***) to evaluate CVCNI. (As we were comparing only 2 conditions, it’s equivalent to independent t-test). We only compared the optimal stimuli for each neuron instead of all the stimuli as in the 4x imaging because single neurons are often very selective, while 4x signals are Gaussian smoothed and are therefore contributed by many nearby cell bodies and neurites. We added the same analysis for CVSI and CNSI in ***Figure 2 — figure supplement 3A-B.*** For CVSI, we compare the maximum response to curves against the maximum response to corners and bars (one-way ANOVA, repeat = 10, p < 0.05), and 70.5% (74 out of 105) neurons with CVSI > 0.2 significantly preferred curves over corners and bars. For CNSI it was 76.9% (120 out of 156).

We also added permutation test to the clustering analysis in Figure 2I-K. For instance, in Figure 2I, tuning correlation and distances were randomly paired for 100,000 times to build the null distribution and average to derive the dash curve. A point was considered significant if it’s higher than the top 100 of the null distribution (p < 0.001) or lower than the bottom 100 (Methods line 556-559). We said in the old version that the curve and corner domains are about 400 μm in size, now we changed it to 300.

2. Limitations were noted about the fine-grained analyses with the 16x objective. A limitation of the present work is that there is only one pair of patches for each animal images at 16x. The analyses need to be also extended.

The possible choice for 16x imaging is quite limited (Figure 1F-G and Figure 1 — figure supplement 3). Yue et al. 2014 reported only one significant curvature patch in dorsal V4, and we would not be surprised if these patches are not too many in V4d. We also have to avoid dense blood vessels in 16x imaging as it will heavily affect the visualization of clustering. In fact, we had another monkey but GCaMP6s was expressed in that monkey. GCaMP6s is easily saturated and hard to differentiate strong and weak responses, and therefore may be not extremely suitable for quantitative and semi-quantitative analysis, though its absolute response can be very strong.

The authors should further analyze the topographic organization at the local scale, is the transition sharp or gradual, what is the variability etc. It seems that there is a rather sharp boundary and that the tuning stays relatively flat, looking at the Figures 2A-D, 3A, 4C and seems particular clear in 5A and C (using concentric versus radial gratings). However, there is no real quantification of this. Figure 2I-K shows the tuning over distance, but this analysis seems to be performed without taking the shape of the domain into account. One possibility would be to show a similar plot, but where the axis is taken perpendicular to the boundary of the domain. It seems that the interpretation that the authors give of the data would predict that the selectivity shows a sharp transition at the boundary and stays elevated within the domain. Furthermore, it would be relevant to get an estimate of the averaged selectivity as well as the variability within the domain, separately for the two animals. Finally, it would be relevant to compare both the sharpness of the transition as well as the mean and variability with the domain between the different stimulus set (curves versus angles, and concentric versus radial gratings).

This is a very good point. We now show the plot over distances to the boundary in Figure 2 — figure supplement 3 and Figure 5 — figure supplement 2. To us the transition is rather gradual. Generally it took around 300 μm before getting elevated. The transition of CRI maps of monkey A in Figure 5A intuitively looked sharp probably because too many neurons had negative CRI values. CRI and CVCNI were correlated but not identical. Since concentric gratings only have 360° full circles but some neurons might prefer short arch (small radian, Figure 3 — figure supplement), it is possible that they don’t respond strongly to concentric gratings. We added these discussions in the revised manuscript (line 313-321).

3. Data visualisationThe authors should show more raw data (high-resolution fluorescence images with the field of view used for the main analyses), as well as traces of fluorescence as a function of time as is standard with imaging to appreciate the quality of the fluorescence traces (over tens of seconds). In addition showing dF/F responses for single neurons to different stimuli would be important.

Yes, we have added the raw fluorescence traces of neurons in Figure 2 — figure supplement 1. We show the responses to the optimal curves, corners and bars together in figure supplement 2.

4. Bar tuningIt needs to be very clear if the small amount of bar tuning reported is only in the ROIs that are defined by subtracting bars (where this would be therefore expected) or overall, in the discussion it currently sounds like this is the case overall which was not clear from the results.

It is only in the FOVs defined in **Figure 1F-G** white box. Sorry we didn’t make it clear. We wrote in the old version L297: “This was not the case for our data, which we infer is primarily due to sampling neurons within or close to curve and corner domains.” We considered it the reason for both the above two points but it unfortunately seemed ambiguous. We have rewritten this part the revised manuscript (line 327-335).

5. Choice of stimuliThe exact choice of stimulus needs to be discussed: why only black (other studies used only white stimuli), why only lines (not surfaces as in e.g. Pasupathy et al. study that is referred to), why no colors. Is it assumed that this will not matter for the results and why?

Bushnell and Pauspathy 2012 study demonstrated shape encoding are largely consistent across colors. So we think the curve/corner preference may be to some extent color invariant and the choice of color may not matter too much. Besides, from our natural image result we did not find dominant color dimension in the curve and corner domains, and the preferred natural images of curves can be of various color. Our V1 study used mostly black lines (Tang et al. 2018), so we just used black lines.

The bar length is matched to the radius of the curve stimuli, which implies to me that the overall number of black pixels is never matched for bars vs the other categories? The authors should discuss if this is a problem.

Yes, but the long bars are precisely matched to the small corners in pixel number, and very closely matched to 60° curves (length ratio = 1:π/3 = 1:1.05). We compare the maximum responses to 60° curves against bars for curve selective neurons in Figure 3D and Figure 3 — figure supplement 1C-D, and we found 52.4% significantly preferred 60° curves to bars, only 7.1% preferred bars to 60° curves (Author response image 2). An example of this is Neuron 3 in Figure 2G. Note that many neurons preferred longer curves and might not even respond to 60° curves (Figure 3 — figure supplement 1D). So we think even with similar pixel number many neurons still preferred curves over bars, and curve preference is not a mere artifact of more pixels.

**Author response image 2. sa2fig2:** Scatterplot showing the maximum response to 60° curves against the maximum response to bars of the curve selective neurons.

Do you expect more curve/corner functional domains if you use different color or luminance contrast, or do you expect the non-significantly curve/corner clustered parts of V4 to contain other functional domains?

We are not exactly sure if more curve/corner domains can be found, but Bushnell and Pauspathy 2012 reported shape tuning remain consistent across color, and from our natural images result the neurons in one curve or corner could respond to images containing curves or corners of various color.

There are certainly other functional domains in V4, such as orientation (Tanigawa et al. 2010), color (Conway et al. 2007), spatial frequency (Lu et al. 2018) and 3D vision (Srinath et al. 2020).

6. Temporal dynamicsThe imaging technique confines analyses to a late time window. If possible refer to literature demonstrating that response preferences remain similar across time for these stimuli, since tuning can be dynamic over time (e.g. Nandy et al. 2016, Issa and DiCarlo eLife).

Yes, the time it takes for Calcium influx and accumulation limits the temporal resolution so that we only record late responses. In fact, it’s known that the early and late responses of V4 neurons could be quite different (Yau et al. 2013). The early responses are considered feed-forward signals and are therefore more tuned to local orientation. Complex pattern preference emerges gradually and is more likely to follow a recurrent model. We are currently undertaking some loose patch recordings in V4 and we find that the early and late responses can be different. The limitation of temporal resolution cannot at present be overcome using Calcium based fluorescence. Hopefully the newly developed neurotransmitter or voltage sensors may help to achieve high temporal resolution imaging. We have added some of this to the discussion in the revised manuscript (line 353-362).

7. Introduction and Discussion:Intro and Discussion read quite well and a lot of the relevant literature is referred to. But Intro and Discussion could include further/more explicit clarification why exactly this contrast (curve/corner) was used to study functional domains (or is this just a starting point), what other functional domains there could be.

We used curve/corner based on the early version of this manuscript posted on bioRxiv in 2019: https://www.biorxiv.org/content/10.1101/808907v2.full (The recent 2 *eLife* papers from Roe and Lu both cited this manuscript). In brief, we recorded V4 neurons’ responses to thousands of natural images, and purely data-driven dimensional reduction on population responses without any apriori assumption. Apart from feature dimensions encoding simple orientation, we also found a dimension encoding curves and corners (on the opposite directions of the axis). The curve and corner selective neurons were separately spatially clustered. However, there are very limited curve and corner domains, and may other functional domains like orientation (Tanigawa et al. 2010), color (Conway et al. 2007) and spatial frequency (Lu et al. 2018) could be found in V4. If the result is to be repeated, one will have to first localize the curve/corner domain and then recording natural images responses (otherwise other feature dimensions might be more dominant), which is against the idea of “purely data-driven without apriori assumption”. So we removed the natural image part and simplify the manuscript to only focus on curve and corner domains in V4. We added some of these to the discussion in the revised manuscript (line 299-302).

The paper needs to cite literature relating curve/corner to animate/inanimate contrasts you discuss (e.g. Zachariou et al. 2018, and other work from Yue lab). You may consider a brief discussion/mention the potential use or function of functional topographic clustering (e.g. Kanwisher, DiCarlo), which is proposed to be related to naturalistic experience that is also discussed here without references.

We have restructured the final paragraph of the discussion to encompass the thread of ideas a little bit more clearly, and we have added references related to the concepts of functional specialization, including its specificity in terms of cognition.

The history presented in the introductory section of this paper is very strange. The first paper that reported curvature tuning in V4 was the Gallant et al. 1993 paper that is cited ambiguously here. It is true that paper used gratings rather than curved lines, but a neuron that is selective for curved gratings is also likely selective for curved lines. A similar principle holds for the hyperbolic grating selectivity reported in Gallant et al. 1993. The authors should address this directly and acknowledge the relationship late in their paper.Similarly, in their subsequent 1996 longer report Gallant et al. argued that neurons selective for curved and hyperbolic gratings were spatially clustered. The data presented in the paper under review is far better than the data that were available to Gallant et al. way back in 1996, but this result was anticipated by that earlier 1996 report, however this finding is not cited.

Yes, we should have directly acknowledged these findings. These studies are now referred to on line 50-53 and line 250-253 of the revised manuscript.

The authors should discuss the recent paper by Roe lab on curvature patches using intrinsic optical imaging has just been published in eLife: https://elifesciences.org/articles/57261. This paper is relevant for relevant the points above, as they claim that there is a smooth transition from rectilinear to low curvature to high curvature (figure 7).The authors should furthermore discuss this recent eLife paper on curvature domains, using both intrinsic and 2-photon imaging: https://elifesciences.org/articles/57502

These papers are now referred to on line 305-311 (both papers cite our 2019 preprint, and were not yet published when we submitted our manuscript to *eLife*). Roe’s paper used curved gratings vs straight gratings, while in our case it’s mainly curve vs corner. The 90° corners and the Π-shapes we used were of comparable level of curvature as smooth curves, but were encoded more in the corner domains. So we feel that our data may not be suitable to address the low curvature to high curvature transition problem.

[Editors' note: further revisions were suggested prior to acceptance, as described below.]

Reviewer #2:The authors replied sufficiently to most of the comments.One answer is not clear to me, in response to the comment:"Some more details about the expression levels would be useful. Most importantly, it is unclear from Figure 1C-F how homogenous the expression was in the selected region. Could you show a separate image where it is possible to judge the level of expression? Also, would it be possible to give an estimate of the general expression levels in terms of percentage of total neurons, as well as the percentage of neurons that were nucleus filled? Finally, it would be relevant to know injection speed in this regard."First of all, they still do not provide the injection speed.

We are sorry for missing this in the previous revision. The injection speed is 5-10 nl/s. We have added this to Materials and methods line 408 of the revised manuscript.

Also, they write: "Most of the neurons that are clearly visible in an average image are not nucleus filled (Figure 1—figure supplement 2)."However, Figure 1—figure supplement 2 does not show any individual cells. Nor do any of the other supplementary figures provide an image where it is possible to judge the structure of the labelling in individual cells, so allowing to see whether they have a clear donut shape, or are nucleus filled. It would therefore still be relevant to see a large / high resolution image where this can be judged.

We firstly apologize for our mistaken figure labelling. Figure 1—figure supplement 2 here should be Figure 2—figure supplement 1A. We have now rearranged the layout of Figure 2—figure supplement 1 in the manuscript to make the image larger and have uploaded the source TIF file in the system. We hope that this revised figure satisfies your concern.

Reviewer #4:The authors have put a lot of work into this revision and the paper is substantially improved over the initial submission. The paper is still largely replicative and confirmatory, but there is a place in the literature for such papers.It is reported that the V4 receptive fields sampled here were very close to the fovea. That implies that the viewing window was very far lateral, much farther than most prior V4 studies. My intuition is that the ear would have had to be removed in order to access V4 at this location. If the authors recorded more medially then I suggest that they recheck their reported eccentricity to be sure that it is correct.

Yes, the optical window is indeed quite lateral. As can be seen in ***Figure 1B***, a large part of the IOS and PIT is included in the 10 mm diameter window, and we were imaging in the lower half. Nevertheless, since the imaging window is smaller than those used for ISOI we did not remove any of the auricle; though the edge of the window was very close to it. In fact, we can also implant such windows to image PIT without removing the auricle.

The indexes that are used here have a pretty unintuitive and unusual scaling range. (For example, an index of 0.2 indicates a 1.5 times difference.) The paper would probably be easier to understand if they had a more intuitive range/form. (For example, if 1.5 indicated a 1.5 times difference.) However, this is up to the authors' discretion.

We agree that the selectivity indexes can be unintuitive. But such a definition is often used to characterize orientation selectivity (OSI, orientation selectivity index), and is also commonly used in some recent imaging studies such as those by Wilson et al. 2018 and Garg et al. 2019. We therefore think this precedent justifies the use such indexes.

Figure 2I "significant" is misspelled. There are also a few places throughout the manuscript where pronouns are missing. (I commend the authors on the English though, it is generally quite good!)

We have corrected the misspelling in Figure 2I, and thank you for your commendation.

Also in Figure 2, please spell out what "CVSI" and "CNSI" mean in the caption. In this and other captions, it is best if the reader can generally understand the caption on its own, w/o having to wade through the text.

We agree this is helpful to the reader. We now spell them out in the captions of Figure 2A-B, as well as Figure 3A and Figure 5A.

The use of hexagonal segments to try to understand differences in tuning for curves versus angles is a weak approach, because hexagonal shapes are a poor intermediate model for these feature classes. A much more powerful method for understanding these differences would be to use an explicit computational model. But that seems to be beyond the scope of this paper…

This is a very valid criticism. We are still working on a computational model to understand how neurons encode curves and corners differently and also to interpret our natural images data. But, as you said, it’s beyond the current scope of this paper. We do hope that we can better address this in our future work.